# Video-Twin Technique for Airway Management, Combining Video-Intubating Stylet with Videolaryngoscope: A Case Series Report and Review of the Literature

**DOI:** 10.3390/healthcare10112175

**Published:** 2022-10-30

**Authors:** Bo-Jyun Jhuang, Hsiang-Ning Luk, Jason Zhensheng Qu, Alan Shikani

**Affiliations:** 1Department of Anesthesia, Hualien Tzuchi Hospital, Hualien 97002, Taiwan; 2Bio-Math Laboratory, Department of Financial Engineering, Providence University, Taichung 43301, Taiwan; 3Department of Anesthesia, Critical Care and Pain Medicine, Massachusetts General Hospital, Harvard Medical School, Boston, MA 02114, USA; 4Division of Otolaryngology-Head and Neck Surgery, LifeBridge Sinai Hospital, Baltimore, MD 21218, USA; 5Division of Otolaryngology-Head and Neck Surgery, MedStar Union Memorial Hospital, Baltimore, MD 21218, USA

**Keywords:** airway management, difficult airway, tracheal intubation, laryngoscopy, videolaryngoscope, video-assisted intubating stylet, anesthesia, COVID-19

## Abstract

Direct laryngoscopy (DL) and videolaryngoscopy (VL) have been the most commonly used airway management modalities in the last several decades. Meanwhile, various optional airway tools (e.g., supraglottic devices, fiberoptic bronchoscopes) have been used for alternative rescue modalities when anticipated or unexpected difficulties in airway management has occurred. In recent decades, optical stylets and video-assisted intubating stylets have become another option for difficult airway scenarios. In contrast to other approaches, we have adopted the Shikani video-assisted intubating stylet technique (VS) for both routine and difficult airway management scenarios. In this case series report, we present the video-twin technique, combining a videolaryngoscope with a video-assisted intubating stylet in various clinical case scenarios. We propose that such a combination is easy to learn and employ and is particularly beneficial in situations where an expected difficult airway (EDA) is encountered.

## 1. Introduction

Conventional direct laryngoscopy (DL) has been the mainstream intubation tool for the last century. In 2010, videolaryngoscopy (VL) shifted the paradigm of airway management from providing direct vision to indirect vision [1]. Comparisons between VL and DL have then been widely studied and the advantages and superiority of the VL over DL have been reported in various clinical scenarios [2,3,4,5,6,7,8]. A seeing-intubating stylet has been invented more recently, along with a video-stylet intubating technique (VS), or the Shikani technique, for intubation [9], and was quickly introduced into various clinical scenarios [10,11,12]. In our hands (Haulien TzuChi Medical Center), it has demonstrated multiple advantages over DL and VL. To the airway operator, VS is available, affordable and portable; it provides easy maneuverability and clear visualization on an LCD screen; it offers a quick learning curve and a feeling of confidence; the stylet is easy to prepare and disinfect. To patients, VS provides rapid intubation time, high first-attempt and overall success rate, less autonomic overstimulation, less airway trauma, and a lower risk of injury.

With a VS, the airway operator uses his/her non-dominant hand (the thumb) to lift up patient’s chin and mandible in order to open the airway (Figure 1A). When necessary, a second airway assistant can help to perform the two-handed jaw thrust while either standing by the side or opposite to the lead airway operator (Figure 1B). In our experience, this has not been often necessary, and a single airway manager can successfully complete the intubation through VS in most cases (>99%). In the rare occasion where an expected difficult airway (EDA) was encountered, combining VS with DL or VL would widely open the oropharynx, lift up the tongue and epiglottis, and provide a better exposure of the glottis, hence greatly enhancing the success of the tracheal intubation. An example of these situations is shown below; the patient’s airway was facilitated with a tongue root holder/elevator (Figure 1C) or a laryngoscope (DL or VL, Figure 1D). There are reports in the literature about VL being superior to DL in some clinical scenarios [13,14].

The idea of using laryngoscopy to facilitate other airway management tools (e.g., to achieve a better glottis visualization) is not new. In numerous reports, either DL or VL has been used to assist fiberoptic bronchoscopy (FOB) in various clinical scenarios [15,16,17,18,19,20,21,22,23,24,25,26,27,28,29,30,31]. DL was also reported in combination with the fiberoptic stylet to assist in clinical and simulated difficult airway cases [32,33,34,35,36,37,38,39,40,41]. 

We have a unique clinical experience with VS in our institutions, both for routine and difficult airways. We also adopted VS a video-twin technique, combining VL with VS in selected cases with EDA. Figure 2 shows such an example in which VL was used as an adjunct to assist the function of VS. Briefly, after the anesthesia depth and neuromuscular block had reached the optimal level for tracheal intubation, the combined technique was started with insertion of the VL blade. The advantages of first use of VL were to open mouth, lift up the tongue, and avoid obstructed or obscure view by the secretions or mucus. The epiglottis was then easily visualized, with or without entire visualization of the glottis. Subsequently, the VS was inserted into oropharyngeal space and easily reached in front of the epiglottis, under the facilitated scope view by the VL. The advantages, among many others, of using VS under such combination conditions include negotiating the limited oropharyngeal space, acquiring a better view around the laryngeal inlet, the maneuverability of the stylet made easy to slip beneath the epiglottis and reach the glottis opening, assuring the railroading endotracheal tube and definitively entering the trachea under the direct vision, etc. In this brief report, we further present and share our clinical experiences of the combined use both the VS and VL video together in various different clinical scenarios. The average intubation time was less than 15 s (for demonstration purposes) and the procedure was smooth.

## 2. Cases Presentation

In this report, several VL and VS were used as follows: (1) UE rigid laryngoscope and TRS video stylet, Zhejiang UE Medical Corp., Taizhou, 317300, China; (2) C-MAC-VS (Video Stylet), KARL STORZ SE & Co. KG, Tuttlingen, Germany; (3) TuoRen Kingtaek Video Intubating stylet, TuoRen, Henan Tuoren Medical Device Co., Xinxiang, Henan, 453401, China; (4) Trachway video Intubating stylet, Markstein Sichtech Medical Corp., Taichung, 407, Taiwan; (5) McGRATH™ MAC Video Laryngoscope, Medtronic, Minneapolis, MN, 55432-5604, USA.

### 2.1. Case 1 (An Elective Surgery)

A 64-year-old man (body height 161 cm, body weight 65 kg, body mass index (BMI) 25.0 kg/m^2^) underwent laparoscopic cholecystectomy and duodenorrhaphy. His past medical history included type II diabetes mellitus, esophageal reflux, and gall stones. Intraoperatively, standard American Society of Anesthesiologists (ASA) monitoring was applied, including non-invasive blood pressure (NIBP), electrocardiogram (ECG), end-tidal CO_2_ (EtCO_2_), pulse oximeter (SpO_2_), and peripheral temperature monitor. General anesthesia was induced with intravenous medications, including lidocaine (20 mg), fentanyl (1 µg/kg), propofol (1.5 mg/kg) and rocuronium (0.6 mg/kg). The tracheal intubation was performed using combined technique with VS and VL. Briefly, the operator inserted a videolaryngoscope with his non-dominant hand (left hand) and localized the glottis. Then, the operator applied video-assisted intubating stylet with his dominant hand. Using this technique, the access of the stylet into the airway was swift and glottis view was clear and full (Figure 3). The endotracheal tube was then pushed into the trachea smoothly and its position was confirmed with EtCO_2_ signals and equal bilateral breath sounds. The intubation time (from lip to trachea) was 8 s. Anesthesia was maintained with sevoflurane, fentanyl, and rocuronium. The whole procedure was uneventful.

### 2.2. Case 2 (Hypopharyngeal Cancer)

A 54-year-old man (175 cm, 59 kg, BMI 19.2 kg/m^2^) was admitted for laryngeal microsurgery (LMS). He was diagnosed with right hypopharyngeal squamous cell carcinoma (SqCC, cT2N0M0) three years ago and completed concurrent chemoradiotherapy (CCRT). This time, he was found to have right aryepiglottic folds (AE fold) granulation and underwent laryngeal biopsy (suspected dysplasia squamous papilloma). American Society of Anesthesiologists standard monitoring was applied. General anesthesia was induced with intravenous glycopyrrolate, lidocaine, fentanyl, midazolam, and propofol. Muscle paralysis was induced by succinylcholine for tracheal intubation (with an endotracheal tube-ID 6.0 mm). The combined VS with VL technique facilitated by videolaryngoscopy was applied. The intubating time (from lip to trachea) was 8 s (Figure 4). Anesthesia was maintained with sevoflurane and cis-atracurium. The whole procedure was smooth.

### 2.3. Case 3 (Cervical Spine Immobility) 

A 44-year-old woman (160 cm, 60 kg, BMI 23.4 kg/m^2^) underwent augmentation enterocystoplasty. Her past medical history included cervical spinal cord injury (C5–C7 injuries with complete paraplegia five years previously), urinary incontinence and frequent urinary tract infection, and autonomic dysreflexia (neurogenic voiding dysfunction and detrusor sphincter dyssynergia). Standard peri-operative vital signs monitorings were performed. General anesthesia was induced with lidocaine, ketamine, propofol, and rocuronium. Anesthesia was maintained with desflurane and cis-atracurium and supplementary fentanyl. Tracheal intubation was performed using the combined VS with VL (Figure 5). Intubation time was 11 s.

### 2.4. Case 4 (Thyroidectomy with Neurofunction Monitoring) 

A 73-year-old woman (151 cm, 72 kg, BMI 31.5 kg/m^2^) underwent total thyroidectomy and right central cervical lymph node dissection due to papillary thyroid carcinoma. Her past medical history included esophageal squamous cell carcinoma (SqCC cT1N0M0, stage 1, received CCRT), arrhythmia, DM, and hypertension. Intraoperative neuromonitoring (IONM) was used for bilateral lymph node exploration and to confirm the functional integrity of the recurrent laryngeal nerve (RLN) as well as facilitates identification of the RLN before visualization during operations. Standard peri-operative vital signs monitoring was performed. General anesthesia was induced with lidocaine, fentanyl, propofol, and rocuronium. Anesthesia was maintained with sevoflurane. Tracheal intubation was performed using the combined VS with VL technique (Figure 6). Intubation time was 9 s.

### 2.5. Case 5 (COVID)

A 71-year-old man (170 cm, 53 kg, BMI 18.3 kg/m^2^) underwent tongue biopsy, trans urethral removal of bladder tumour (TURBT), ureterorenoscopic manupilation (URS-M), and insertion of double-J stents. The patient was diagnosed with chronic obstructive pulmonary disease (COPD), urinary bladder urothelial cancer (high-grade UBUC, cT2N0M0) and tongue cancer. General anesthesia was induced with lidocaine, glycopyrrolate, fentanyl, propofol and rocuronium. He was diagnosed with COVID-19 with positive PCR results. The anesthesia staff were equipped with full-gear personal protective equipment (PPE). Anesthesia was maintained with sevoflurane. Tracheal intubation was performed using the combined VS with VL technique (Figure 7). Intubation time was 6 s.

### 2.6. Case 6 (Morbid Obesity)

A 42-year-old man (169 cm, 120 kg, BMI 42.0 kg/m^2^) underwent left percutaneous nephrolithotomy (PCNL) and left ureterorenoscope stone manipulation (URS-SM). Standard peri-operative vital signs monitors were applied. General anesthesia was induced with lidocaine, glycopyrrolate, fentanyl, midazolam, propofol and rocuronium. Anesthesia was maintained with sevoflurane. Tracheal intubation was performed using the combined VS with VL technique (Figure 8). Intubation time was 10 s.

## 3. Discussion

This case series shows the usefulness of the video-twin technique (combined use of VL and VS) for tracheal intubation in six cases (Table 1). While both DL and VL are the mainstream of the intubating tools worldwide, the modern roles of Shikani VS has recently gained in popularity in some regions, especially in Asia. According to our own clinical experiences, VS has demonstrated some advantages including the ease of the intubation process (e.g., intubation time, first-attempt, and overall success rates), safety (fewer airway injuries, hypoxemia, and autonomic over-stimulation), and operators’ subjective satisfaction (e.g., easiness to handle and reduced mental load). Similar results have been observed in manikin studies [42,43,44].

Although from our unique clinical experiences the Shikani VS technique seems to be superior to DL or VL, there are nevertheless some potential technical obstacles that still need to be overcome, especially for novice trainees. Similar to other optical devices, airway secretions, mucus, saliva or blood may obstruct the view of the lenses or chip at the end of the stylet. In addition, the airway soft tissues and structures may collapse, block the view, and leave little space for the stylet to advance. Unless an EDA is encountered, it has been our experience that most elective airway intubations can be successfully accomplished by a single airway operator. It is imperative to clearly understand potentially difficult airways and how to prepare for the optimal working environment for intubation and decide whether to use VS or the video-twin technique combining VS with VL. During nasotracheal intubation using fiberoptic bronchoscopy (FOB), it was found that DL performed almost as well as an adjunct maneuver as jaw thrust for laryngeal visualization and intubation time, and it provided significantly better airway clearance at the level of the soft palate (instead of the larynx) than jaw thrust [15]. The usefulness of combining FOB and VL has been reported in difficult airways, such as those with a huge goiter [26,28,45].

The authors have had clinical experience with the Shikani VS technique for routine tracheal intubation since 2016 [9,46,47,48,49,50,51,52,53]. In our 1100-bed medical center (7000 tracheal intubations annually), VS technique has been routinely applied in more than 90% of the patients who received tracheal intubation. The rest of the tracheal intubations were performed with DL and VL mainly for teaching purposes. Here we propose the combined use of VS with VL (the video-twin technique) for tracheal intubation. The VL laryngoscope is held with the non-dominant hand, the laryngoscope blade sits in the vallecula widely opening the oropharyngeal space, lifting the tongue, and gently elevating the epiglottis (which is particularly helpful in cases of a droopy/floppy epiglottis), providing an easier visualization of the vocal cords with a smooth railroading of the stylet through the glottis. Among many other advantages, the advantages of such combination method include allowing a one-operator technique without the need for a second assistant (Table 2). The prices of DL (from 100 to 1500 USD), VL (from 1600 to 7000 USD), and VS (from 200 to 8000 USD) varied by different manufacturers and in different regions. It is worth mentioning a few shortcomings of our proposed video-twin technique, including the need for a longer time to set up the two systems, more costs to equip an extra VL or VS, competition for the same oral space by the two instruments, and a potentially longer learning curve for novice learners, etc.

In most cases, the classic Shikani VS technique using one hand to lift chin is manageable and easily done by a single airway operator [9]. Occasionally, adopting a two-handed jaw thrust maneuver helps to further elevate the chin, optimizing access to the airway. However, a second airway assistant would be required on the scene [54,55]. It has been reported that using a tongue holder/retractor helps when performing an FOB procedure [56]. We have also found that using a tongue holder/retractor may help to open a patient’s oropharyngeal space when VS is used. In comparison to a simple tongue retractor device, a VL laryngoscope blade provides a better opportunity to visualize the airway and lower the chances of accidental injuries (e.g., Figure 2). In conclusion, we have found that the video-twin technique (combined use of VS and VL) may offer some advantages over VS alone, especially in cases of difficult airways.

## 4. Examples of Expected Difficult Airways (EDAs)

High body mass index, severe obstructive sleep apnea;Severe head and neck trauma, tumor, restricted cervical spine motion;Narrow oropharynx (huge tongue, significant micrognathia, Mallampati 4);Very anterior larynx, Cormack-Lehane grade 3/4, droopy/floppy epiglottis that prevent the scope from seeing the glottis;Bleeding, heavy secretions, vomits (obstructs the view of the stylet and requires continuous suctioning).

## Figures and Tables

**Figure 1 healthcare-10-02175-f001:**
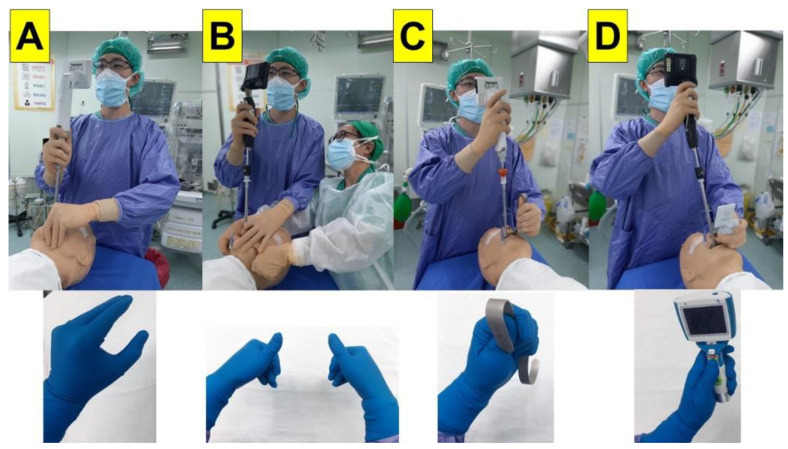
Typical maneuvers of using video-assisted intubating stylet technique: (**A**) classic Shikani technique. The airway operator uses his/her non-dominant hand to perform jaw-thrust; (**B**) two-handed jaw-thrust technique. The second airway operator assists opening of the airway and elevates the jaw; (**C**) use of tongue root holder/elevator to open the airway and access to the larynx; and (**D**) use of a videolaryngoscope either by the same operator or an assistant airway manager. Intubating devices: Trachway (**A**,**C**); C-MAC-VS (**B**,**D**); McGRATH™ MAC (**D**, upper panel); UE (**D**, lower panel).

**Figure 2 healthcare-10-02175-f002:**
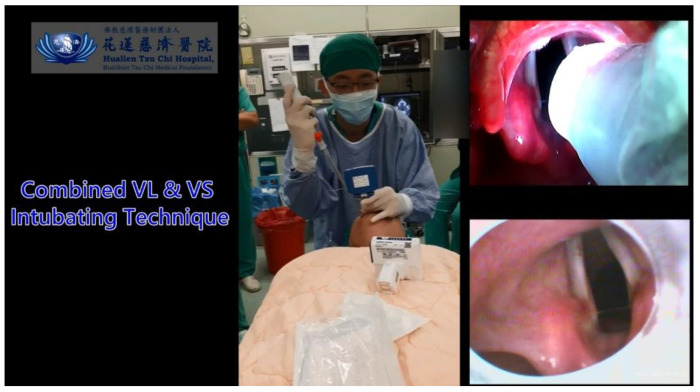
A demonstration video clip showing the video-twins technique. A 24-year-old man with BMI 21.4 kg/m^2^ (183 cm, 72 kg). The airway operator holds a videolaryngoscope in his non-dominant hand and maneuvers a video-assisted intubating stylet in his dominant hand. The Cormack-Lehane class I, POGO 100%, and intubation time by intubating stylet is 15 s (from lip to trachea). (Also, see the Appendix A). Intubating devices: Trachway and UE.

**Figure 3 healthcare-10-02175-f003:**
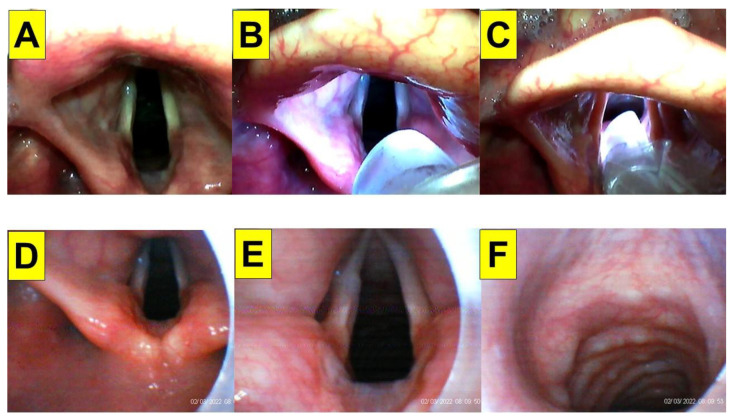
Application of video-twins technique in a patient underwent an elective surgery. A 64-year-old man with a BMI 25.0 kg/m^2^. Mallampati class III and Cormack-Lehane laryngoscopic view grade 1: (**A**–**C**) videolaryngoscopic views; and (**D**–**F**) views from intubating stylet. Intubation time was 8 s. (also see the Appendix A). Intubating devices: TuoRen Kingtaek and Trachway.

**Figure 4 healthcare-10-02175-f004:**
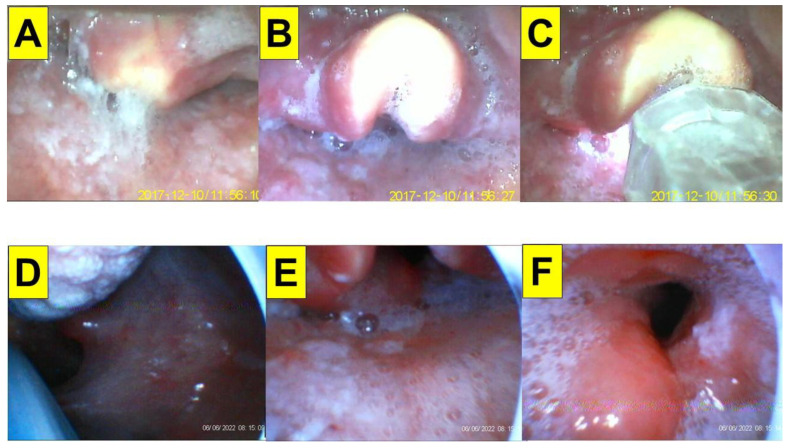
Application of video-twins technique in a patient underwent an elective surgery. A 54-year-old man with a BMI 19.2 kg/m^2^. Mallampati class III and Cormack-Lehane laryngoscopic view grade 3: (**A**–**C**) videolaryngoscopic views; and (**D**–**F**) views from intubating stylet. Intubation time was 8 s. (also see the Appendix A). Intubating devices: UE and Trachway.

**Figure 5 healthcare-10-02175-f005:**
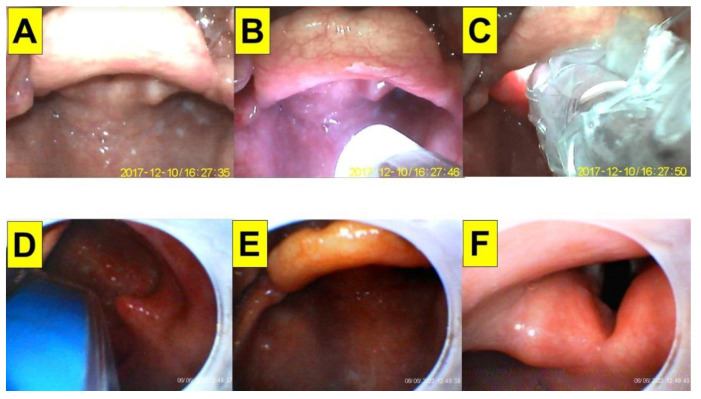
Application of video-twins technique in a patient with cervical spine injury. A 44-year-old woman with a BMI 23.4 kg/m^2^. Mallampati class III and Cormack-Lehane laryngoscopic view grade 2b: (**A**–**C**) videolaryngoscopic views; and (**D**–**F**) views from intubating stylet. Intubation time was 11 s. (also see the Appendix A). Intubating devices: UE and Trachway.

**Figure 6 healthcare-10-02175-f006:**
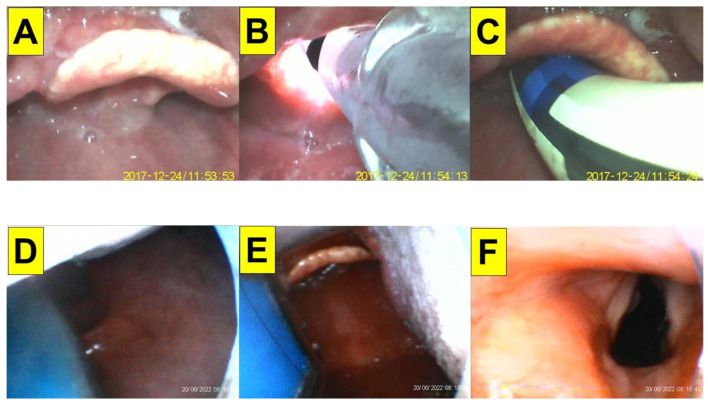
Application of video-twins technique in a patient underwent thyroidectomy. A 73-year-old woman with a BMI 31.5 kg/m^2^. Intraoperative neuromonitoring (IONM) was used. Mallampati class III and Cormack-Lehane laryngoscopic view grade 2b: (**A**–**C**) videolaryngoscopic views.; and (**D**–**F**) views from intubating stylet. The proper position of the IONM tube over the vocal cords was confirmed (**C**). Intubation time was 9 s. (also see the Appendix A). Intubating devices: UE and Trachway.

**Figure 7 healthcare-10-02175-f007:**
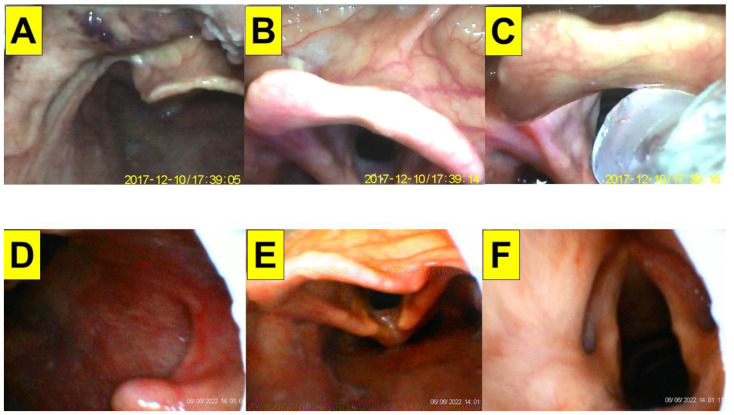
Application of video-twins technique in a patient (a 71-year-old man with a BMI 18.3 kg/m^2^) contracted COVID-19 underwent tongue biopsy and genitourinary surgeries. Mallampati class was not evaluated and Cormack-Lehane laryngoscopic view was grade 2a: (**A**–**C**) videolaryngoscopic views; and (**D**–**F**) views from intubating stylet. The proper position of the IONM tube over the vocal cords was confirmed (**C**). Intubation time was 6 s. (also see the Appendix A). Intubating devices: UE and Trachway.

**Figure 8 healthcare-10-02175-f008:**
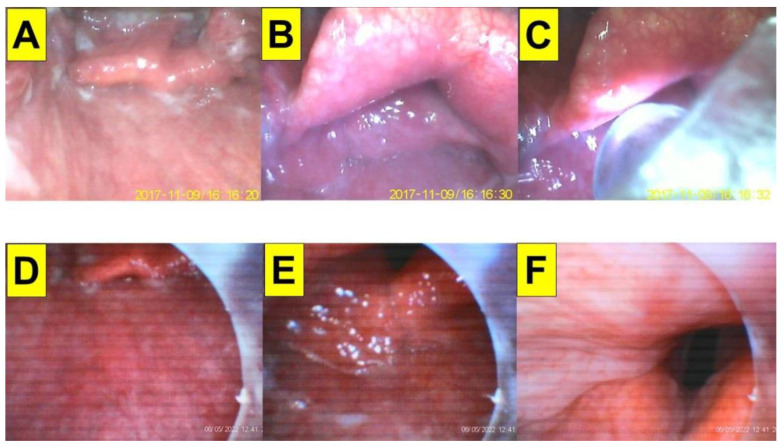
Application of video-twins technique in a morbidly obese patient (a 42-year-old man with a BMI 42.0 kg/m^2^) underwent genitourinary surgeries. Mallampati class III and Cormack-Lehane laryngoscopic view was grade 2b: (**A**–**C**) videolaryngoscopic views; and (**D**–**F**) views from intubating stylet. Intubation time was 10 s. (also see the Appendix A). Intubating devices: UE and Trachway.

**Table 1 healthcare-10-02175-t001:** Summary of the six cases.

	Case 1	Case 2	Case 3	Case 4	Case 5	Case 6
Age/gender	64/M	54/M	44/F	73/F	71/M	42/M
BMI (kg/m^2^)	25.0	19.2	23.4	31.5	18.3	42.0
Induction	propofol, fentanyl, rocuronium	propofol, midazolam, fentanyl, succinylcholine	propofol, ketamine, rocuronium	propofol, fentanyl, rocuronium	propofol, fentanyl, rocuronium	propofol, fentanyl, rocuronium
Maintenance	sevoflurane	sevoflurane	desflurane	sevoflurane	sevoflurane	sevoflurane
Airway evaluation	MMT:II C-L:1 ULBT:1 SMD:14 cm NC:38 cm	MMT:III C-L:3 ULBT:2 SMD:12 cm NC:33 cm	MMT:III C-L:2b ULBT:1 SMD:15 cm NC:35 cm (C-spine immobility)	MMT:III C-L:2b ULBT:1 SMD:12 cm NC:40 cm	MMT:NA C-L:2a ULBT:NA SMD:NA NC:NA (COVID-19 positive)	MMT:III C-L:1 ULBT:1 SMD:12 NC:45
Surgery	laparoscopic cholecystectomy	laryngomicrosurgery	augmentation enterocystoplasty	thyroidectomy (with IONM)	removal of bladder tumor, tongue biopsy	percutaneous nephrolithotomy
Indication of video-twin technique	Relative	Relative	Relative	Absolute	Relative	Relative
First-pass success	Yes	Yes	Yes	Yes	Yes	Yes
Intubation time (VS)	8 s	8 s	11 s	9 s	6 s	10 s
Complications	Nil	Nil	Nil	Nil	Nil	Nil

M/F: male/female; BMI:body mass index; MMT: modified Mallampati test; C-L: Cormack-Lehane grade; ULBT: upper lip bite test; SMD: sterno-mental distance; NC: neck circumference; IONM: intra-operative neuromonitoring; intubation time: from lip to trachea by VS (video stylet).

**Table 2 healthcare-10-02175-t002:** Qualitative comparison of various tracheal intubation modalities.

	DL	VL	VS	VL + VS
Difficult laryngoscopy	**++**	**+**	**-**	**+**
Impeded visualization by the soft tissues	**-**	**-**	**+++**	**-**
Obtain high-grade C-L/POGO score	**±**	**++**	**++** (requires jaw thrust)	**+++**
High first-pass success rate	**-**	**+**	**+++**	**+++**
Disturbed by saliva, secretions, etc	**-**	**-**	**+++** (requires suction)	**±**
Easy intubation	**±**	**+**	**++**	**+++**
ET tube insertion hits glottis structures	**+++**	**++**	**-**	**-**
Identify proper ET tube insertion depth	**+**	**++**	**-**	**++**
Post-operative sore throat	**+++**	**++**	**-**	**+**
Dental/soft tissue injuries	**+++**	**++**	**-**	**+**
High cost-affordability	**-**	**+++**	**++**	**++**
Learning barrier for novice airway operator	**+++**	**++**	**+**	**+**

DL: direct laryngoscope; VL: videolaryngoscope; VS: video-assisted intubating stylet; C-L: Cormack-Lehane grading; POGO: the percentage of glottic opening scale; ET tube: endotracheal tube.“+”: yes; “-”: no; “±”: inconclusive.

## Data Availability

Not applicable.

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
