# Peer review of "Video-Twin Technique for Airway Management, Combining Video-Intubating Stylet with Videolaryngoscope: A Case Series Report and Review of the Literature"

_healthcare, 2022, doi:10.3390/healthcare10112175_

Round 1

Reviewer 1 Report

excellent article. May want to discuss this technique for smaller size infants

Author Response

MS No: Healthcare-1934297

MS Title: Video-twin technique…..

MS Authors: Jhung et al.

Reviewer-2

Comment-1: Excellent article. May want to discuss this technique for smaller size infants.

Response-1: We can only make a short remark here about the potential role of the video-twin technique in smaller-size infants. This is because it is not necessary to apply such combining technique in small infants (e.g., smaller mouth opening, already not difficult to apply smaller size of the VS for such pediatric population, etc). Intuitively, the simple VS technique is perhaps good enough for tracheal intubation in smaller infants scenario. Most important, we have not started systemically trying VS technique in pediatric population. This is due to the ethics consideration. We have to first demonstrate the validity and feasibility of the VS technique in the adult populations before we can test such assumption in the most vulnerable human subject population.

Thanks for your excellent comment!

Reviewer 2 Report

Thank you for asking me to review the article "Video-twin technique for airway management, combining video-intubating stylet with videolaryngoscope".

Combined use of two video techniques for "normal" and difficult intubation is an interesting concept.

In this article, eight patient's findings are presented. 

I am not sure that the authors can persuade clinicians to use this technique in everyday practise. 

The authors should discuss the cost of this technique.

Authors' suggestions and definition of unexpected difficult airway is wrong, as at least first three categories are not unexpected. They can simply say that it should be available for difficult airway and that sometimes they should use this technique for the training purpose. If the technique is so good, it will work in unexpected difficult airway situations, that are very rare. 

The authors measure time of tracheal intubation in seconds, but they do not compare with "normal intubation" time. 

The algorithm that they suggest is bot based on any evidence and does not seem to contribute to better outcome when there is a difficult intubation. 

The authors did not mention any shortfall of the technique and/or possible complications.

Minor comments are:

Introduction, lines 41-43: Is there any reference for that statements or it is the authors' assumption?

The same question is for the lines 184-190 in the discussion.

Lines 191-196: What are these conclusions based on?

The sentence: The authors have vast clinical experience" should be replaced with number of cases or the length of time during which they used the technique.

Author Response

MS No: Healthcare-1934297

MS Title: Video-twin technique…..

MS Authors: Jhung et al.

Reviewer-2

The authors would like to thank the reviewer for the excellent comments and constructive suggestions.

Comment 1: Thank you for asking me to review the article "Video-twin technique for airway management, combining video-intubating stylet with videolaryngoscope". Combined use of two video techniques for "normal" and difficult intubation is an interesting concept. In this article, eight patient's findings are presented. I am not sure that the authors can persuade clinicians to use this technique in everyday practice.

Response-1: Thanks for your excellent comments. To apply video-assisted intubating stylet technique (VS) and/or the video-twin technique is indeed an “avante-garde” concept for airway management community. Since Dr. Alan Shikani brought in the idea of a “seeing” intubating technique in 1999, there is a trend of applying such excellent intubation option on difficult airway scenarios, and now as we demonstrated in the last decade, on “routine” airway management. We understand the residual impact of both the conventional direct laryngoscopy and videolaryngoscopy (also kind of a new paradigm) in the real-world 21th century. However, we have tremendous advancement in using VS technique on “routine” tracheal intubation on daily basis since 2016. For example, in our 1100-bed medical center in Taiwan (Hualien Tzuchi Medical Center), nearly 7000 tracheal intubations a year, more than 90% of these cases were intubated by VS technique. And the rest of the tracheal intubations either by videolaryngoscopy or direct laryngoscopy were mainly for teaching purpose. Therefore, before the airway management community accepts VS as the transcending role for tracheal intubation, the novice practitioners might find the ease to use videolaryngoscope (or direct laryngoscope) on the non-dominant hand and VS on the dominant hand. This video-twin technique serves a kind of bridging practice to move forward from DL/VL laryngoscopy to universal use of VS on routine basis (when the second airway assistant is not available on the scene). Most interesting, we found in very rare difficult airway scenarios, the video-twin technique is in particular a valuable rescue option. That is exactly the purpose of our report to present and to persuade the airway managers this alternative thought (tracheal intubation on daily practice).

Comment-2: The authors should discuss the cost of this technique.

Response-2: The extra cost of such video-twin is the VS. The discussion of the cost of this technique is briefly mentioned now.

The added new text:

“The prices of DL (from 100 to 1500 USD), VL (from 1600 to 7000 USD), and VS (from 200 to 8000 USD) varied by different manufacturers and in different regions.”

Comment-3: Authors' suggestions and definition of unexpected difficult airway is wrong, as at least first three categories are not unexpected. They can simply say that it should be available for difficult airway and that sometimes they should use this technique for the training purpose. If the technique is so good, it will work in unexpected difficult airway situations, that are very rare.

Response-3: Thanks for your excellent correction. We totally agree that those scenarios (e.g., obesity, OSAS, head/neck abnormal structures or lesions etc.)  do not belong to the categories of the “unexpected” DA. On the contrary, we regard (or predict) those conditions to be “expected or anticipated” potential DA and should handle with vigilance. We therefore revised the term and some issues of this Expected Difficulty Airway (EDA).

In addition, we delete the proposed algorithm in the original text in order not to cause unnecessary confusion and misunderstanding.

Comment-4: The authors measure time of tracheal intubation in seconds, but they do not compare with "normal intubation" time.

Response-4: In literature, one can find the intubation time varied among different laryngoscopy. When compared VL against DL, e.g., the intubation time ranged from seconds to minute (Analysis 1.7, page 516, Videolaryngoscopy versus direct laryngoscopy for adults undergoing tracheal intubation. Hansel et al., Cochrane Database Syst Rev. 2022 Apr 4;4(4):CD011136. doi: 10.1002/14651858.CD011136.pub3.) . Unfortunately, it is not yet possible to pool data for time required for tracheal intubation owing to considerable heterogeneity in this Cochrane review. It is also noted that the common definition of intubation time is different between DL/VL and VS. The former counts the time from insertion of laryngoscopic blade into mouth (or hands-on the intubating tool to the airway operator) to complete acquisition of confirmed capnography. In contrast, for VS, we use different definition for the intubation time. We count the intubation time “from touching the lip to the moment of entry into trachea-seeing the tracheal rings”. It is always checked by the playback of the video-recording. Normally, with sole VS technique, the intubation time ranges from 3 sec to 10 sec. Therefore, it was not our intention to “compare” the conventional intubation time among DL, VL, and VS. In this report, we only present our intubation time using VS/VL technique only for a reference (please see Table 1).

Comment-5: The algorithm that they suggest is not based on any evidence and does not seem to contribute to better outcome when there is a difficult intubation.

Response-5: Thanks for your excellent point! We agree with your comment. Therefore, in order not to cause unnecessary confusion or misunderstanding, we deleted the content regarding the “algorithm” issue.

Comment-6: The authors did not mention any shortfall of the technique and/or possible complications.

Response-6: The shortcomings of the video-twin technique has now added in the revised text.

Page Line 216, added new text:

“Among many others, the advantages of such combination method include allowing a one-operator technique without the need for second assistant (Table 2). It is worthy to mention few shortcomings of our proposed video-twin technique, including need longer time to set up the two systems, more costs to equip an extra VL or VS, competing for the same oral space by the two instruments, potential longer learning curve, etc.”

Comment-7: Introduction, lines 41-43: Is there any reference for that statements or it is the authors' assumption?

Response-7:

Regarding the statement in the Page 1, lines 41-43, “To patients, VS provides rapid intubation time, high first-attempt and overall success rate, less autonomic overstimulation, less airway trauma and a lower risk of injury.”:

  • As stated in line 38, “In our hands” specified the following statement is based on our own clinical use experiences of the VS technique, which is tremendous in our medical center (Hualien Tzuchi Medical Center) since 2016.
  • Please also see the revised sentences regarding the following Comment/Response-10: “The authors have had clinical experience with the Shikani VS technique for routine tracheal intubation since 2016 [9, 41-48]. In our medical center (7000 intubations annually), VS technique has been routinely applied in more than 90% of the patients. The rest of intubations were performed with DL and VL mainly for teaching purpose.”
  • It is worthy to mention that the prospective head-to-head clinical comparison between VL and VS is still not feasible due to the stringent ethical consideration for human subject protection.

Comment-8: The same question is for the lines 184-190 in the discussion.

Response-8: Page lines 184-190, “While both DL and VL are the mainstream of the intubating tools worldwide, the modern roles of Shikani VS has recently gained its popularity in some regions, especially in Asia. VS has demonstrated some advantages including the ease of the intubation process (e.g., intubation time, first-attempt and overall success rates), safety (less airway injuries, hypoxemia, autonomic over-stimulation), and operator’s subjective satisfaction (e.g., easiness to handle, reduced mental load).” The same explanation as revised in the Page 1 and Page 8, the statements are purely based on our unique clinical experiences in Hualien TzuChi Medical Center.

Comment-9: Lines 191-196: What are these conclusions based on?

Response-9: Page 7, Lines 191-196 “Although from our clinical experiences the Shikani VS technique seems to be superior to DL or VL, there are nevertheless some technical obstacles still need to overcome, especially for novice trainees. Similar to other optical devices, airway secretions, mucus, saliva or blood may obstruct the view of the lenses or chip at the end of the stylet. In addition, the airway soft tissues and structures may collapse, block the view, and leave little space for the stylet to advance.

Again, such statement is purely based on our unique clinical experiences of using the technique.

Comment-10: The sentence: “The authors have vast clinical experience” should be replaced with number of cases or the length of time during which they used the technique.

Response-10: Page 8, Line 207, “The authors have a vast clinical experience with the Shikani VS technique [9, 41-48].” has now been revised as follows.

“The authors have had clinical experience with the Shikani VS technique for routine tracheal intubation since 2016 [9, 41-48]. In our medical center (7000 intubations annually), VS technique has been routinely applied in more than 90% of the patients. The rest of intubations were performed with DL and VL mainly for teaching purpose.”

Reviewer 3 Report

I would like to thank the authors for their work. The purpose of this article is clearly stated. However, there are various concerns and criticisms regarding the manuscript, which are itemized below.

1.     Title: I suggest this one: Video-twin technique for airway management, combining video-intubating stylet with a video laryngoscope. A case series and review of the literature.

2.     P1L43: add a reference

3.     Figures: The authors should indicate the devices and the manufacturer of all devices that appear in this case series.

4.     Cases: a more practical presentation for the reader of the cases could be:

·      Add this section: An Overview of the Technique. It should include the advantages of the technique. It is well explained in the article: Gómez-Ríos MA, Nieto Serradilla L. Combined use of an Airtraq® optical laryngoscope, Airtraq video camera, Airtraq wireless monitor, and a fibreoptic bronchoscope after failed tracheal intubation. Can J Anaesth. 2011 Apr;58(4):411-2.

·      Replace the description of the cases with a summary table (case 1-case 6). Include, in addition to demographic characteristics, anesthetic induction, anesthesia maintenance, the characteristics of each airway (test), the indication of the technique in each case (the reason why it was used)

·      In airways with tumours that received radiotherapy treatment, they sometimes present friable mucosa. Why they opted for this technique and not for a fiberoptic tracheal intubation.

 5.     Discussion.

·      P7L190: add a reference

·      I suggest that the authors add a table with different case reports published and different techniques combined

·      The algorithm is confusing. It would be more clarify an algorithm as a figure instead of a simple text.

Good luck

Author Response

MS No: Healthcare-1934297

MS Title: Video-twin technique…..

MS Authors: Jhung et al.

Reviewer-3

The authors wish to thank the reviewer for his/her excellent comments and constructive suggestions. We responded and made revisions as follows:

I would like to thank the authors for their work. The purpose of this article is clearly stated. However, there are various concerns and criticisms regarding the manuscript, which are itemized below.

Comment-1: Title: I suggest this one: Video-twin technique for airway management, combining video-intubating stylet with a video laryngoscope. A case series and review of the literature.

Response-1: The title has been revised accordingly.

“Video-twin technique for airway management, combining video-intubating stylet with a video laryngoscope. A case series report and review of the literature.”

Comment-2: P1L43: add a reference

Response-2: Page 1, Lines 38-43: regarding the reference to the statement…

  • “In our hands (Haulien TzuChi Medical Center),” has been added in the revised text in order to emphasize the following statement is based on our unique clinical experiences of using the VS technique.
  • Also this statement has been coined by a revised text in the Page 8. “The authors have had clinical experience with the Shikani VS technique for routine tracheal intubation since 2016 [9, 41-48]. In our 1100-bed medical center (7000 tracheal intubations annually), VS technique has been routinely applied in more than 90% of the patients who received tracheal intubation. The rest of intubations were performed with DL and VL mainly for teaching purpose.”

Comment-3:  Figures: The authors should indicate the devices and the manufacturer of all devices that appear in this case series.

Response-3:

Excellent point! The following relevant information has been revised and added into the text.

  • “In this report, several VL and VS have been used as follows: (1) UE rigid laryngoscope and TRS video stylet, Zhejiang UE Medical Corp., Taizhou, 317300, China; (2) C-MAC-VS (Video Stylet), KARL STORZ SE & Co. KG, Tuttlingen, Germany; (3) Tuoren Kingtaek Video Intubating stylet, TuoRen, Henan Tuoren Medical Device Co., Xinxiang, Henan, 453401, China; (4) Trachway video Intubating stylet, Markstein Sichtech Medical Corp., Taichung, 407, Taiwan; (5) McGRATH™ MAC Video Laryngoscope, Medtronic, Minneapolis, MN, 55432-5604, USA.”

Comment-4: Cases: a more practical presentation for the reader of the cases could be:

[1] Add this section: An Overview of the Technique. It should include the advantages of the technique. It is well explained in the article: Gómez-Ríos MA, Nieto Serradilla L. Combined use of an Airtraq® optical laryngoscope, Airtraq video camera, Airtraq wireless monitor, and a fibreoptic bronchoscope after failed tracheal intubation. Can J Anaesth. 2011 Apr;58(4):411-2.

[2] Replace the description of the cases with a summary table (case 1-case 6). Include, in addition to demographic characteristics, anesthetic induction, anesthesia maintenance, the characteristics of each airway (test), the indication of the technique in each case (the reason why it was used)

[3] In airways with tumours that received radiotherapy treatment, they sometimes present friable mucosa. Why they opted for this technique and not for a fiberoptic tracheal intubation.

Response-4:  

[1] An overview of the technique has been added.

  • “Briefly, after the anesthesia depth and neuromuscular block had reached the optimal level for tracheal intubation, the combined technique was started with insertion of the VL blade. The advantages of first use of VL were to open mouth, lift up the tongue, and avoid obstructed or obscure view by the secretions or mucus. The epiglottis was then easily visualized, with or without entire visualization of the glottis. Subsequently, the VS was inserted into oropharyngeal space and easily reached in front of the epiglottis, under the facilitated scope view by the VL. The advantages, among many others, of using VS under such condition include negotiating the limited oropharyngeal space, acquiring a better view around the laryngeal inlet, the maneuverability of the stylet made easy to slip under the epiglottis and reach the glottis opening, assuring the railroading endotracheal tube entering trachea definitively under the direct vision, etc.”

The new citation has been made:

  1. Gómez-Ríos, M.A.; Nieto Serradilla, L. Combined use of an Airtraq® optical laryngoscope, Airtraq video camera, Airtraq wireless monitor, and a fibreoptic bronchoscope after failed tracheal intubation. Can J Anaesth. 2011, 58, 411-412. doi: 10.1007/s12630-011-9460-3.

[2] We added an additional table to provide all the necessary information.

Table 1: Summary of the six cases

Case 1

Case 2

Case 3

Case 4

Case 5

Case 6

Age/gender

64/M

54/M

44/F

73/F

71/M

42/M

BMI (kg/m2)

25.0

19.2

23.4

31.5

18.3

42.0

Induction

propofol,

fentanyl,

rocuronium

propofol,

midazolam,

fentanyl,

succinylcholine

propofol,

ketamine,

rocuronium

propofol,

fentanyl,

rocuronium

propofol,

fentanyl,

rocuronium

propofol,

fentanyl,

rocuronium

Maintenance

sevoflurane

sevoflurane

desflurane

sevoflurane

sevoflurane

sevoflurane

Airway evaluation

MMT:II

C-L:1

ULBT:1

SMD:14 cm

NC:38 cm 

MMT:III

C-L:3

ULBT:2

SMD:12 cm

NC:33 cm

MMT:III

C-L:2b

ULBT:1

SMD:15 cm

NC:35 cm

(C-spine immobility)

MMT:III

C-L:2b

ULBT:1

SMD:12 cm

NC:40 cm

MMT:NA

C-L:2a

ULBT:NA

SMD:NA

NC:NA

(COVID-19 positive)

MMT:III

C-L:1

ULBT:1

SMD:12

NC:45 

Surgery

laparoscopic cholecystectomy

laryngomicrosurgery

augmentation enterocystoplasty

thyroidectomy (with IONM)

removal of bladder tumor, tongue biopsy

percutaneous nephrolithotomy

Indication of Video-twin technique

Relative

Relative

Relative

Absolute

Relative

Relative

First-pass success

Yes

Yes

Yes

Yes

Yes

Yes

Intubation time (VS)

8 s

8 s

11 s

9 s

6 s

10 s

Complications

Nil

Nil

Nil

Nil

Nil

Nil

M/F:male/female; BMI:body mass index;MMT: modified Mallampati test;C-L:Cormack-Lehane grade;ULBT:upper lip bite test;SMD:sterno-mental distance;NC:neck circumference;IONM:intra-operative neuromonitoring;intubation time:from lip to trachea by VS

[3] Regarding the concerns about friable mucosa in those head/neck cancer patients pre/post radiotherapy, it is indeed a dilemma of use or not use FOB for tracheal intubation in this patient population. We totally agree that FOB is the gold standard on this scenario. However, one also has to face the fact on using FOB regarding the friable mucosa (e.g., throughout the nasopharyngeal path) and still facing the risk of bleeding and mucosa damage. More troublesome, one might meet the condition of impingement of the ET tube on the glottis structures and difficulty railroading the ET tube over FOB into the trachea (please see the following references).

  • Touré T, Williams SR, Kerouch M, Ruel M. Patient factors associated with difficult flexible bronchoscopic intubation under general anesthesia: a prospective observational study. Can J Anaesth. 2020 Jun;67(6):706-714. English. doi: 10.1007/s12630-020-01568-w. Epub 2020 Jan 17. PMID: 31953669.
  • Asai T, Murao K, Johmura S, Shingu K. Effect of cricoid pressure on the ease of fibrescope-aided tracheal intubation. Anaesthesia. 2002 Sep;57(9):909-13. doi: 10.1046/j.1365-2044.2002.02706.x.
  • Katsnelson T,  Frost  EA,  Farcon  E,  Goldiner    When  the endotracheal  tube  will  not  pass  over  the  flexible  fiberoptic bronchoscope. Anesthesiology 1992; 76: 151-2.
  • Jackson AH, Orr B, Yeo C, Parker C, Craven R, Greenberg SL. Multiple sites of impingement of a tracheal tube as it is advanced over a fibreoptic bronchoscope or tracheal tube introducer in anaesthetized, paralysed patients. Anaesth Intensive Care. 2006 Aug;34(4):444-9. doi: 10.1177/0310057X0603400409. PMID: 16913339.
  • Marfin AG, Iqbal R, Mihm F, Popat MT, Scott SH, Pandit JJ. Determination of the site of tracheal tube impingement during nasotracheal fibreoptic intubation. Anaesthesia. 2006 Jul;61(7):646-50. doi: 10.1111/j.1365-2044.2006.04652.x. PMID: 16792609.
  • Asai T, Shingu K. Difficulty in advancing a tracheal tube over a fibreoptic bronchoscope: incidence, causes and solutions. Br J Anaesth. 2004 Jun;92(6):870-81. doi: 10.1093/bja/aeh136. Epub 2004 Apr 30. PMID: 15121723.

Therefore, if the conditions allowed, we preferred to try first the VS technique in order to acquire better pharyngeal/laryngeal views and easier railroading the ET tube over the VS. In this case, the VS technique caused less chance of the soft tissue damage or bleeding.

Comment-5: Discussion.

[1] P7L190: add a reference

[2] I suggest that the authors add a table with different case reports published and different techniques combined

[3] The algorithm is confusing. It would be more clarify an algorithm as a figure instead of a simple text.

Response-5:

[1] Page 7, Line 190: Here we added three relevant articles for references regarding the advantages of the VS.

  1. Tseng, K.Y.; Chau, S.W.; Su, M.P.; Shih, C.K.; Lu, I.C.; Cheng, K.I. A comparison of Trachway intubating stylet and Airway Scope for tracheal intubation by novice operators: a manikin study. Kaohsiung J Med Sci. 2012, 28, 448-451. doi: 10.1016/j.kjms.2012.02.016.
  2. Hung, K.C.; Tan, P.H.; Lin, V.C.; Wang, H.K.; Chen, H.S. A comparison of the Trachway intubating stylet and the Macintosh laryngoscope in tracheal intubation: a manikin study. J Anesth. 2013, 27, 205-210. doi: 10.1007/s00540-012-1491-6.
  3. Ong, J.; Lee, C.L.; Huang, S.J.; Shyr, M.H. Comparison between the Trachway video intubating stylet and Macintosh laryngoscope in four simulated difficult tracheal intubations: A manikin study. Ci Ji Yi Xue Za Zhi. 2016, 28,109-112. doi: 10.1016/j.tcmj.2016.06.004.

[2] Since we have introduced numerous references in the Introduction section (references from  to ), we believe an extra table for this citation summary will further lengthen the contents of this manuscript. However, we do add two more references for this notion.

  1. Matioc, A.A. Use of the Airtraq with a fibreoptic bronchoscope in a difficult intubation outside the operating room. Can J Anaesth. 2008, 55, 561-562. doi: 10.1007/BF03016679.
  2. Gómez-Ríos, M.A.; Nieto Serradilla, L. Combined use of an Airtraq® optical laryngoscope, Airtraq video camera, Airtraq wireless monitor, and a fibreoptic bronchoscope after failed tracheal intubation. Can J Anaesth. 2011, 58, 411-412. doi: 10.1007/s12630-011-9460-3.

[3] Thank you for your excellent correction. We then delete the “algorithm” in order not to cause unnecessary confusion or misunderstanding.

Round 2

Reviewer 3 Report

The manuscript has been improved. Good work.

Please, add the specific device also in the figure legends.

L 236: Change this sentence "in this case series report, we demonstrate the video-twin technique  (combined use of VL and VS) for tracheal intubation in six cases (Table 1)" to "This case series could show the usefulness of the video-twin technique (combined use of VL and VS)."

Author Response

MS No: Healthcare-1934297

MS Title: Video-twin technique…..

MS Authors: Jhung et al.

20221024

Comment-1: Please, add the specific device also in the figure legends.

Response-1: The names of the specific devices have now been added in each figure legends (together with the detailed info of the manufacturers in the Page 3, Lines 103-110). Thanks.

Comment-2: L 236: Change this sentence "in this case series report, we demonstrate the video-twin technique (combined use of VL and VS) for tracheal intubation in six cases (Table 1)" to "This case series could show the usefulness of the video-twin technique (combined use of VL and VS)."

Response-2: The new sentence has been revised as follows: " This case series could show the usefulness of the video-twin technique (combined use of VL and VS) for tracheal intubation in six cases (Table 1). "  Note: Here to introduce the Table 1.

Thank you very much for your correction.
